# Mechanistic insights into robust cardiac $I_{Ks}$ potassium channel activation by aromatic polyunsaturated fatty acid analogues

Briana M Bohannon[1†], Jessica J Jowais[1†], Leif Nyberg[1,2], Vanessa Olivier-Meo[1], Valentina Corradi[3], D Peter Tieleman[3], Sara I Liin[2], H Peter Larsson[1]*

[1]Department of Physiology and Biophysics, Miller School of Medicine, University of Miami, Miami, United States; [2]Department of Biomedical and Clinical Sciences, Linköping University, Linköping, Sweden; [3]Department of Biological Sciences and Centre for Molecular Simulation, University of Calgary, Calgary, Canada

*For correspondence: plarsson@med.miami.edu

[†]These authors contributed equally to this work

**Abstract** Voltage-gated potassium ($K_V$) channels are important regulators of cellular excitability and control action potential repolarization in the heart and brain. $K_V$ channel mutations lead to disordered cellular excitability. Loss-of-function mutations, for example, result in membrane hyper-excitability, a characteristic of epilepsy and cardiac arrhythmias. Interventions intended to restore $K_V$ channel function have strong therapeutic potential in such disorders. Polyunsaturated fatty acids (PUFAs) and PUFA analogues comprise a class of $K_V$ channel activators with potential applications in the treatment of arrhythmogenic disorders such as long QT syndrome (LQTS). LQTS is caused by a loss-of-function of the cardiac $I_{Ks}$ channel – a tetrameric potassium channel complex formed by $K_V7.1$ and associated KCNE1 protein subunits. We have discovered a set of aromatic PUFA analogues that produce robust activation of the cardiac $I_{Ks}$ channel, and a unique feature of these PUFA analogues is an aromatic, tyrosine head group. We determine the mechanisms through which tyrosine PUFA analogues exert strong activating effects on the $I_{Ks}$ channel by generating modified aromatic head groups designed to probe cation–pi interactions, hydrogen bonding, and ionic interactions. We found that tyrosine PUFA analogues do not activate the $I_{Ks}$ channel through cation–pi interactions, but instead do so through a combination of hydrogen bonding and ionic interactions.

## Editor's evaluation

This work reports important findings regarding the regulation of ion channels by polyunsaturated fatty acids (PUFAs) through the identification of novel aromatic PUFA analogs with potent effects on the IKs channels, which allow for mechanistic insights into their mode of action. The experiments are solid, combining site-directed mutagenesis, electrophysiological and pharmacological approaches to dissect the different molecular mechanisms and sites involved in the functional interactions. This work will be of broad interest to ion channel biophysicists, physiologists, and medical chemists interested in drug development for LQT syndrome.

## Introduction

The delayed rectifier potassium channel ($I_{Ks}$) underlies a critical repolarizing current that determines the timing of the ventricular action potential (*Nerbonne and Kass, 2005*). The cardiac $I_{Ks}$ current is mediated by the association of the voltage-gated $K^+$ channel $K_V7.1$ α-subunit with the KCNE1

β-subunit (*Abramochkin et al., 2018*; *Wang et al., 1998*; *Barhanin et al., 1996*). The $K_V7.1$ α-subunit consists of six transmembrane spanning segments, denoted S1–S6 where S1–S4 form the voltage-sensing domain (VSD) and S5–S6 form the pore domain (PD) (*Sun and MacKinnon, 2017*). The S4 segment contains several positively charged arginine residues that allow S4 to move outward, toward the extracellular side of the membrane, when the membrane becomes depolarized (*Broomand et al., 2003*). This outward movement of the S4 is transformed into pore opening as a result of conformational changes in the S4–S5 linker of $K_V7.1$ (*Kalstrup and Blunck, 2018*). Co-expression of KCNE1 with $K_V7.1$ imparts a more depolarized voltage dependence of activation, slower activation kinetics, and increased single-channel conductance compared to $K_V7.1$ alone (*Sun et al., 2012*; *Yang and Sigworth, 1998*). Loss-of-function mutations in the cardiac $I_{Ks}$ channel can lead to an arrhythmogenic disorder known as long QT syndrome (LQTS), which predisposes individuals to ventricular fibrillation and sudden cardiac death (*Fernández-Falgueras et al., 2017*; *Roden, 2008*; *Watanabe et al., 2005*). Current treatments for LQTS include pharmacological intervention with β-blockers or surgical implantation of a cardioverter defibrillator (*Waddell-Smith and Skinner, 2016*). However, limitations of these treatments generate a need for novel therapeutic interventions to treat LQTS.

Polyunsaturated fatty acids (PUFAs) are amphipathic molecules composed of a charged hydrophilic head group and a long, polyunsaturated hydrophobic tail group (*Jump, 2002*). It is well documented that PUFAs form a group of $I_{Ks}$ channel activators that interact with the channel VSD, thus influencing the voltage dependence of $I_{Ks}$ channel activation (*Moreno et al., 2015*; *Moreno et al., 2012*; *Elinder and Liin, 2017*). The ability of PUFAs to activate the $I_{Ks}$ channel makes them great candidates for potential LQTS therapeutics. PUFAs promote $I_{Ks}$ channel activation through an electrostatic interaction between the negative charge of the hydrophilic PUFA head and positively charged arginine residues in the S4 segment of the $I_{Ks}$ channel (*Elinder and Liin, 2017*; *Börjesson and Elinder, 2011*; *Larsson et al., 2020*; *Börjesson et al., 2008*). This electrostatic activation of the $I_{Ks}$ channel is seen as a leftward shift in the voltage dependence of $I_{Ks}$ channel activation that leads to increases in $I_{Ks}$ current. Recently, it has been reported that PUFAs increase $I_{Ks}$ current through two independent effects: one on S4 (as described above) and one on the PD through an electrostatic interaction with a positively charged lysine residue located in S6 (K326) (*Yazdi et al., 2021*; *Liin et al., 2018*). This electrostatic interaction with the K326 mediates an increase in the maximal conductance ($G_{max}$) of the $I_{Ks}$ channel (*Yazdi et al., 2021*; *Liin et al., 2018*). The mechanism through which the negatively charged PUFA head group interacts with positive charges of S4 and S6 is called the lipoelectric hypothesis where the polyunsaturated tail of PUFAs and PUFA analogues incorporates into the cell membrane via hydrophobic interactions and electrostatically attracts the outermost gating charges of S4 as well as positively charged K326 in the S6 segment (*Börjesson et al., 2008*; *Yazdi et al., 2021*; *Bohannon et al., 2019*; *Bohannon et al., 2020*).

PUFA analogues that have the most robust effects on increasing $I_{Ks}$ current are those that have a low pKa value and thus possess a negatively charged head group at physiological pH (*Bohannon et al., 2020*). Examples include PUFAs with glycine or taurine head groups that possess either a carboxyl or sulfonyl head group, respectively (*Bohannon et al., 2020*; *Liin et al., 2016*). We have observed that another PUFA analogue, N-(α-linolenoyl) tyrosine (NALT), has robust effects on $I_{Ks}$ current. NALT is unique in that it possesses a large aromatic tyrosine head group rather than a carboxyl or sulfonyl group present in most of the PUFAs and PUFA analogues that we have characterized. NALT induces a potent leftward shift in the voltage dependence of $I_{Ks}$ channel activation and an increase in the maximal channel conductance, thus increasing overall $I_{Ks}$ current. Here, we aim to determine the mechanism behind $I_{Ks}$ activation by NALT using PUFA analogues with aromatic and modified aromatic head groups. Better understanding of the mechanism of how these aromatic PUFA analogues have more potent $I_{Ks}$ channel activation effects could help to aid future drug development for LQTS.

## Results

### Diverse PUFA analogues with a tyrosine head group activate the $I_{Ks}$ channel

To measure the effects of the aromatic PUFA analogues on the cardiac $I_{Ks}$ channel, we expressed the $I_{Ks}$ channel complex in *Xenopus laevis* oocytes (*Figure 1A*). We co-injected mRNA for the $K_V7.1$ α-subunit and the KCNE1 β-subunit to achieve expression of tetrameric $I_{Ks}$ channels. Using two-electrode

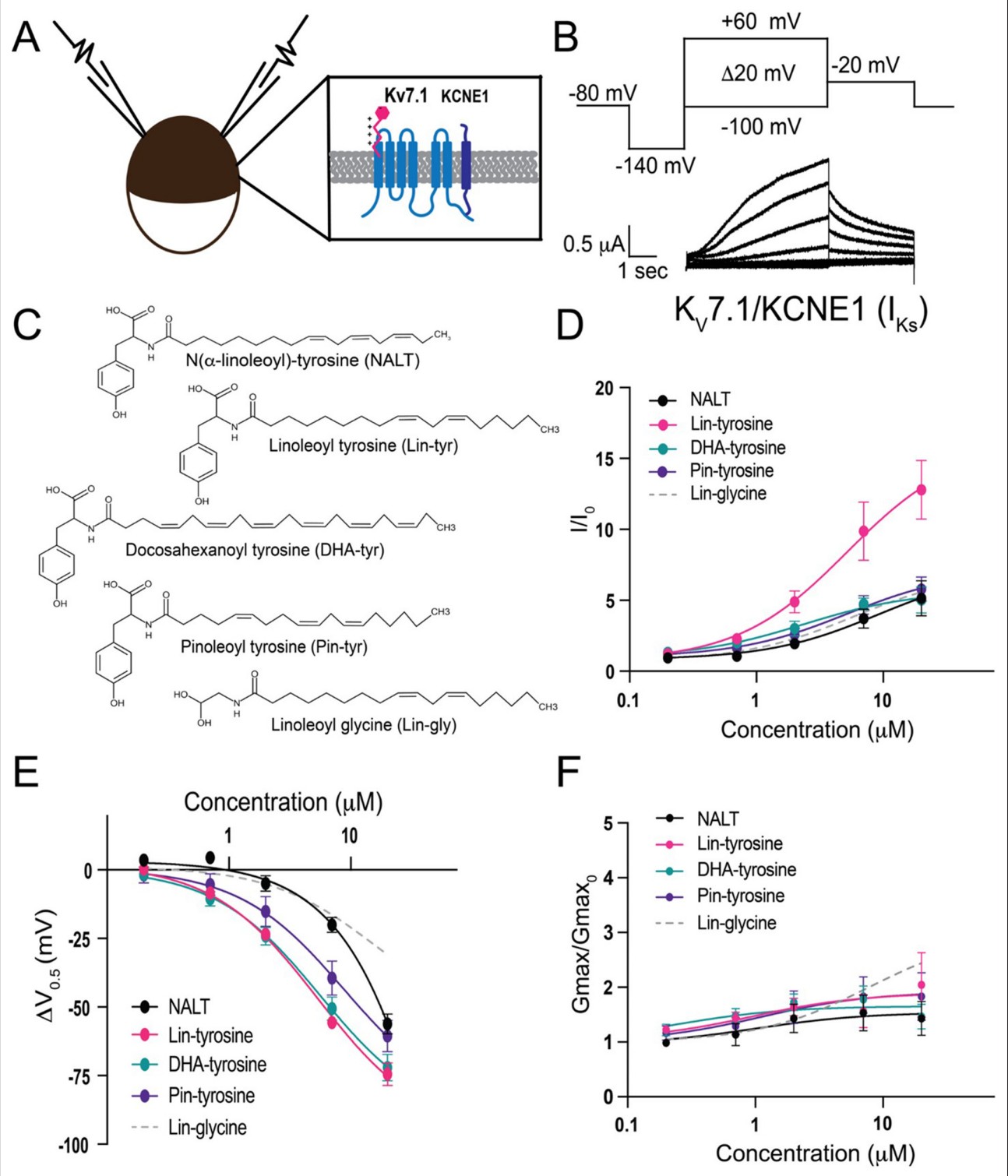

**Figure 1.** Polyunsaturated fatty acids (PUFA) analogues with a tyrosine head group are strong $I_{Ks}$ channel activators. (**A**) Schematic of two electrode voltage-clamp setup (inset: $I_{Ks}$ channel cartoon + PUFA [pink]). (**B**) Voltage protocol (top) with representative $K_V7.1$/KCNE1 ($I_{Ks}$) current (bottom). (**C**) Structures of NALT, Lin-tyrosine, DHA-tyrosine, and Pin-tyrosine (with Lin-glycine for comparison). (**D–F**) $I/I_0$, (**E**) $\Delta V_{0.5}$, and (**F**) $G_{max}$ dose–response curves for NALT (black circles) (n = 4), Lin-tyrosine (pink circles) (n = 4), DHA-tyrosine (teal circles) (n = 3), Pin-tyrosine (purple circles) (n = 5), and Lin-glycine (gray dotted line) (n = 3). Values for all compounds and concentrations available in *Figure 1—source data 1*.

The online version of this article includes the following source data for figure 1:

**Source data 1.** Source Data for Panels D-F.

voltage-clamp recordings, we applied depolarizing voltage steps to activate the $I_{Ks}$ channel (*Figure 1B*) before and after applying four different tyrosine PUFA analogues: N(α-linolenoyl)-tyrosine (NALT), linoleoyl tyrosine (Lin-tyrosine), docosahexaenoyl tyrosine (DHA-tyrosine), and pinoleoyl tyrosine (Pin-tyrosine) (*Figure 1C*). From these voltage-clamp experiments, we are also able to acquire dose–response curves for different aspects of $I_{Ks}$ channel activation, including changes in overall $I_{Ks}$ current ($I/I_0$, *Figure 1D*), changes in the voltage dependence of activation ($\Delta V_{0.5}$, *Figure 1E*), and changes in the maximal channel conductance ($Gmax/Gmax_0$, *Figure 1F*). NALT, Lin-tyr, DHA-tyr, and Pin-tyr all activate the cardiac $I_{Ks}$ channel by shifting the voltage dependence of $I_{Ks}$ channel activation to more negative voltages (NALT: $\Delta V_{0.5} = -56.2 \pm 3.6$ mV; Lin-tyr: $-74.4 \pm 4.1$ mV; DHA-tyr: $-72.0 \pm 4.9$ mV; and Pin-tyr: $-60.5 \pm 5.8$ mV at 20 µM; *Figure 1E*) and increasing the maximal conductance (NALT: $1.43 \pm 0.3$; Lin-tyr: $2.0 \pm 0.6$; DHA-tyr: $1.5 \pm 0.2$; and Pin-tyr: $1.8 \pm 0.4$ at 20 µM; *Figure 1F*). Together, the left shift in $V_{0.5}$ and the increase in $G_{max}$ increase the overall $I_{Ks}$ current measured in response to a voltage step close to 0 mV (NALT: $5.14 \pm 1.2$; Lin-tyr: $12.8 \pm 2.1$; DHA-tyr: $5.0 \pm 0.9$; and Pin-tyr: $5.8 \pm 0.9$ at 20 µM; *Figure 1D*: see 'Materials and methods' for calculation of $I/I_0$). In comparison, Lin-glycine (a known $I_{Ks}$ channel activator; *Figure 1C*) causes only modest leftward voltage shifts ($\Delta V_{0.5} = -30.8 \pm 5.4$ mV at 20 µM; *Figure 1E*), but similar increases in maximal conductance ($2.6 \pm 0.5$ at 20 µM; *Figure 1F*) and $I/I_0$ current ($6.7 \pm 1.1$ at 20 µM; *Figure 1D*) as for most tyrosine PUFAs.

## Distal –OH group is necessary for robust activation of the $I_{Ks}$ channel

Amino acids with aromatic groups (like tryptophan, tyrosine, and phenylalanine) can participate in cation–pi interactions (*Dougherty, 2007*). Cation–pi interactions take place between the pi-electrons of an aromatic ring and positively charged (cationic) groups (such as arginine and lysine) (*Infield et al., 2021*). If tyrosine PUFAs activate the $I_{Ks}$ channel via cation–pi interactions, we would expect that other aromatic groups (such as phenylalanine) would similarly affect $I_{Ks}$ activation. We tested two different PUFA analogues that both contain a phenylalanine head group – linoleoyl phenylalanine (Lin-phe) and N-(α-linolenoyl) phenylalanine (NAL-phe) (*Figure 2A*). Lin-phe and NAL-phe both increase $I/I_0$ (Lin phe: $2.6 \pm 0.3$; and NAL-phe: $2.4 \pm 0.5$ at 20 µM; *Figure 2B–D*), causing a modest leftward shift in the $V_{0.5}$ (Lin-phe: $\Delta V_{0.5} = -13.1 \pm 2.9$ mV; and NAL-phe: $-12.5 \pm 3.8$ mV at 20 µM; *Figure 2E and F*). However, Lin-phe and NAL-phe have minimal effects on the $G_{max}$ (Lin phe: $1.2 \pm 0.1$; and NAL-phe: $1.2 \pm 0.2$ at 20 µM; *Figure 2G and H*). All of these effects ($I/I_0$, $\Delta V_{0.5}$, and $G_{max}$) are reduced in comparison with tyrosine PUFAs, with Lin-phe and NAL-phe having an attenuated effect compared to NALT (p=0.51 and p=0.44, respectively; *Figure 2D*) and resulting in a significantly smaller increases in $I/I_0$ compared to Lin-tyrosine (\*\*\*p=0.0004 and \*\*\*p=0.0004, respectively; *Figure 2D*). In addition, both NALT and Lin-tyrosine cause a significantly greater $\Delta V_{0.5}$ compared to NAL-phe and Lin-phe (\*\*\*\*p<0.0001; *Figure 2F*). Together, these differences suggest that cation–pi interactions are not the primary mechanism through which tyrosine PUFAs activate the $I_{Ks}$ channel. Rather, our data suggest that it is actually the presence of the distal –OH group on the aromatic head group that is critical for the potent activation of the $I_{Ks}$ channel because the loss of this –OH group (Lin-phe and NAL-phe) results in pronounced reductions in PUFA efficacy.

## Electronegative groups on aromatic ring are important for increases in maximal conductance

Our data thus far indicates that it is the presence of the –OH group, not cation–pi interactions, that is critical for pronounced $I_{Ks}$ channel activation by tyrosine PUFAs. The –OH group found in tyrosine PUFAs is highly electronegative. To test how electronegativity influences $I_{Ks}$ channel activation, we compared three modified phenylalanine PUFAs, which all include a highly electronegative group(s) attached to the aromatic ring. We compared N-(α-linolenoyl)–4-bromo-L-phenylalanine (4Br-NAL-phe), N-(α-linolenoyl)–4-fluoro-L-phenylalanine (4F-NAL-phe), and N-(α-linolenoyl) 3,4,5-trifluorophenylalanine (3,4,5F-NAL-phe) (*Figure 3A*). 4Br-NAL-phe, 4F-NAL-phe, and 3,4,5F-NAL-phe application increases $I/I_0$ (4Br-NAL-phe: $4.6 \pm 0.1$ at 20 µM; 4F-NAL-phe: $4.6 \pm 1.1$ at 20 µM; 3,4,5F-NAL-phe: $7.1 \pm 1.0$ at 20 µM; *Figure 3B–D*), causes a leftward shift in the $V_{0.5}$ (4Br-NAL-phe: $\Delta V_{0.5} = -22.8 \pm 2.0$ mV; 4F-NAL-phe: $-23.9 \pm 0.8$ mV; 3,4,5F-NAL-phe: $-32.4 \pm 4.9$ mV at 20 µM; *Figure 3E and F*) and increases the $G_{max}$ (4Br-NAL-phe: $1.9 \pm 0.1$ at 20 µM; 4F-NAL-phe: $2.0 \pm 0.5$; 3,4,5F-NAL-phe: $2.4 \pm 0.4$ at 20 µM; *Figure 3G and H*). Increasing the number of highly electronegative groups significantly improves the effects of phenylalanine PUFAs on increasing $I/I_0$ and shifting the $V_{0.5}$, evidenced by

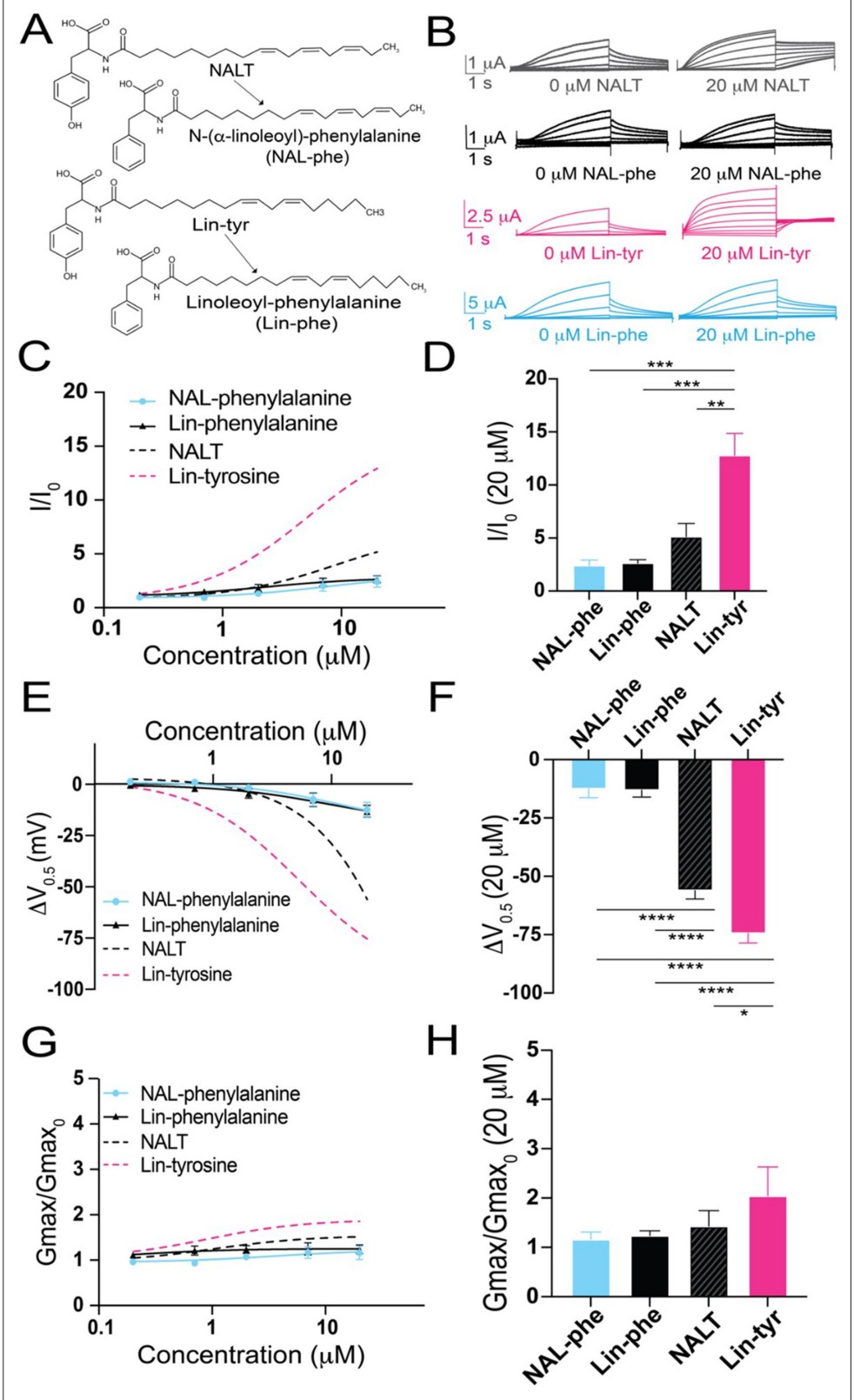

**Figure 2.** The distal hydroxyl (–OH) group of tyrosine polyunsaturated fatty acid (PUFA) analogues is necessary for robust $I_{Ks}$ channel activation. (**A**) Structures of NAL-phe and Lin-phe. (**B**) Representative current traces for NALT (gray), NAL-phe (black), Lin-tyr (pink), and Lin-phe with 0 µM PUFA (left) and 20 µM PUFA (right). (**C, E, G**) $I/I_0$, (**E**) $\Delta V_{0.5}$, and (**G**) $G_{max}$ dose–response curves for NAL-phe (n = 4) and Lin-phe (n = 4) with dotted lines representing

*Figure 2 continued on next page*

*Figure 2 continued*

dose response of NALT (n = 4) and Lin-tyr (n = 4). (**D, F, H**) Maximum effects on (**D**) $I/I_0$, (**F**) $\Delta V_{0.5}$, and (**H**) $G_{max}$ (at 20 µM) for NAL-phe (n = 4), Lin-phe (n = 4), NALT (n = 4), and Lin-tyr (n = 4). Asterisks indicate statistically significant differences determined by one-way ANOVA with Tukey's test for multiple comparisons. Values for all compounds and concentrations available in *Figure 2—source data 1*.

The online version of this article includes the following source data for figure 2:

**Source data 1.** Source Data for Panels C-H.

significant increases in $I/I_0$ (*p=0.0186; *Figure 3D*) and a significantly greater leftward shift in the $V_{0.5}$ (**p=0.0096; *Figure 3F*) from 3,4,5F-NAL-phe compared to NAL-phe alone. Interestingly, though, NALT still causes the most prominent left-shift in the $V_{0.5}$ compared to 4Br-, 4F-, and 3,4,5F-NAL-phe (***p=0.0003; ***P=0.00028; and **p=0.0021, respectively). These data suggest that the presence of highly electronegative groups improve the activating effects of phenylalanine PUFAs on the $I_{Ks}$ channel. However, they do not completely recapitulate the effects of tyrosine PUFAs on the shift in $V_{0.5}$ of the $I_{Ks}$ channel.

## Hydrogen bonding is important for pronounced leftward shifts in $I_{Ks}$ channel voltage dependence

The presence of the –OH group on tyrosine PUFA analogues or the addition of electronegative groups to the phenylalanine head group improves $I_{Ks}$ activation. However, a persistent and striking difference between tyrosine PUFAs and modified phenylalanine PUFAs in the magnitude of their voltage-shifting effects with the tyrosine PUFAs having an almost twice as big voltage shift effect than the modified phenylalanine PUFAs (*Figure 3E and F*). One explanation for this discrepancy is that the –OH group can also behave as a hydrogen bond donor. To determine whether hydrogen bonding contributes to the activating effects of tyrosine PUFA analogues, we applied the modified aromatic PUFA analogue N-(α-linolenoyl)–3-fluoro-L-tyrosine (3F-NALT), which has a fluorine atom adjacent to the tyrosine hydroxyl group (*Figure 4A*). The addition of the fluorine atom reduces the $pK_a$ of the distal hydroxyl group and increases the hydrogen bonding ability of said group in 3F-NALT as compared to NALT. Overall, the maximum effects on $I/I_0$ are similar for 3F-NALT and NALT (3F-NALT: 5.0 ± 1.0; NALT: 5.14 ± 1.2 at 20 µM; p=0.7257, ns; *Figure 4B–D*). Notably, 3F-NALT induces a significantly greater maximum shift in the $V_{0.5}$ ($\Delta V_{0.5}$ = –69.3 ± 1.4 at 20 µM) compared to NALT (–56.1 ± 3.6 AT 20 µM) (p=0.0298*; *Figure 4E and F*), while the effects on $G_{max}$ are not significantly different between 3F-NALT and NALT (3F-NALT: 1.3 ± 0.3; NALT: 1.4 ± 0.3 at 20 µM; p=0.7324, ns; *Figure 4G and H*). These data demonstrate that increasing the hydrogen bonding capacity of the –OH group increases the maximum shift in $I_{Ks}$ channel voltage dependence. This implicates hydrogen bonding as an important mechanism for $I_{Ks}$ activation and preferentially influences the effects on the voltage dependence of $I_{Ks}$ activation.

## Aromatic PUFAs appear to activate the $I_{Ks}$ channel in similar mechanisms as non-aromatic PUFAs

To better understand the mechanism of these more effective aromatic PUFAs, we mutated residues previously shown to be important for non-aromatic PUFA activating effects on $I_{Ks}$ channels. The residue R231, located in the voltage sensor (S4) (*Figure 5A*), has been previously shown to be important for the $V_{0.5}$ shifting effect of non-aromatic PUFAs (*Liin et al., 2018*). We tested Lin-tyr, the largest $V_{0.5}$ shifting aromatic PUFA, on the $I_{Ks}$ channel with the mutation R231Q+Q234R to assess whether R231 is also important for the aromatic PUFA $V_{0.5}$ shifting mechanism. The additional mutation Q234R is necessary to preserve the voltage dependence of activation in $I_{Ks}$ channels with the R231Q mutation (*Liin et al., 2018*; *Panaghie and Abbott, 2007*; *Wu et al., 2010*). The $V_{0.5}$ shifting effect of Lin-tyr was significantly decreased from $\Delta V_{0.5}$ = –74.4 mV ± 4.1 at 20 µM in the wild-type (WT) $I_{Ks}$ channel to $\Delta V_{0.5}$ = –36.5 mV ± 7.3 at 20 µM with the R231Q+Q234R mutation (****p < 0.0001; *Figure 5B and C*). This reduction indicates that R231 contributes to more than half of the voltage dependence shifting effect of Lin-tyr. The remaining shift is most likely due to PUFA head group interactions with other nearby S4 charges such as R228 and Q234R.

The residue K326, located near the pore, has been previously shown to be important for the $G_{max}$ increasing effect of non-aromatic PUFAs (*Liin et al., 2018*). We tested 3,4,5F NAL-phe, the largest

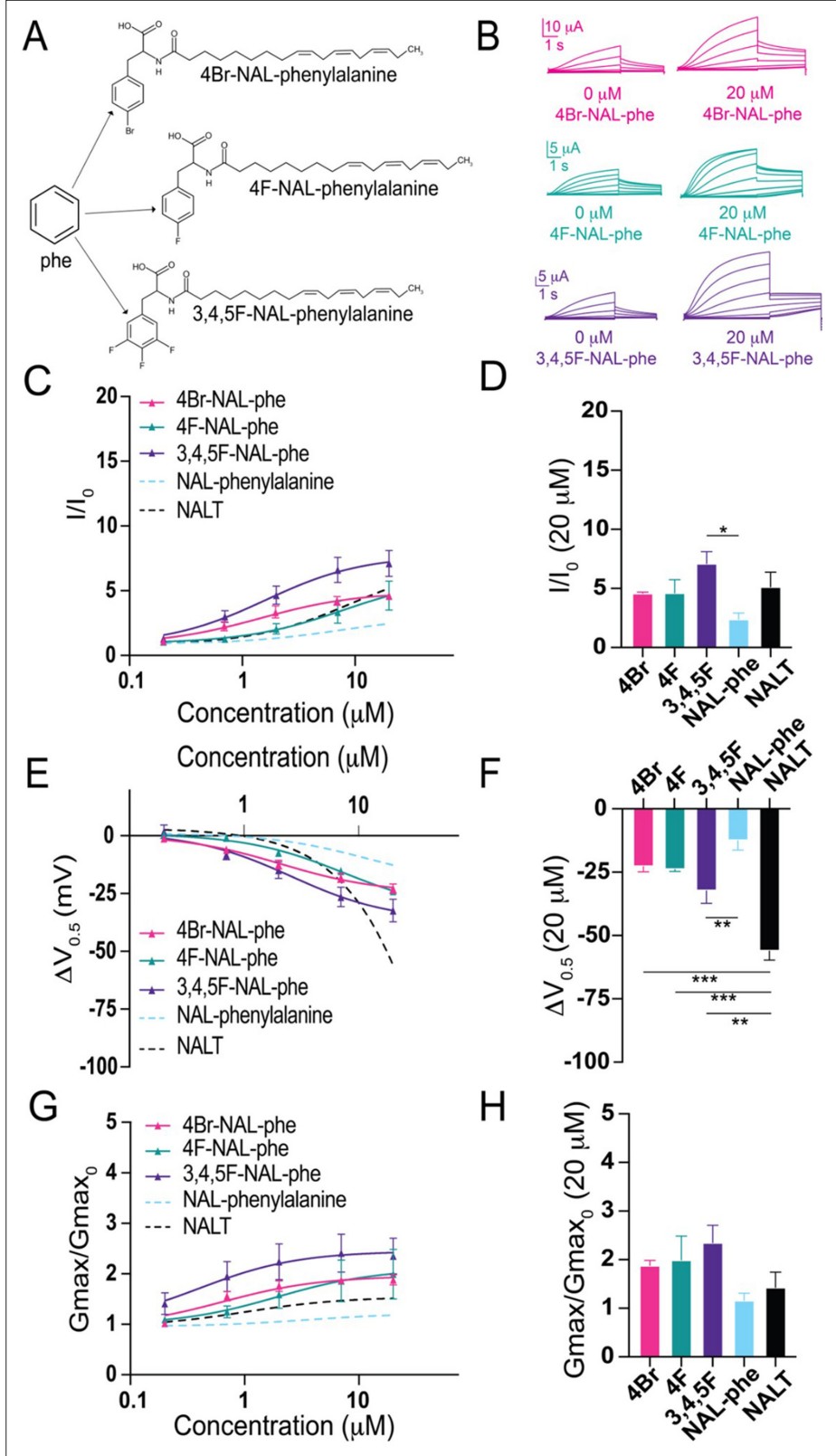

**Figure 3.** The addition of electronegative atoms to phenylalanine polyunsaturated fatty acid (PUFA) analogues strengthens $I_{Ks}$ channel activation through improved effects on $G_{max}$. (**A**) Structures of 4Br-NAL-phe, 4F-NAL-phe, and 3,4,5F-NAL-phe. (**B**) Representative traces for 4Br-NAL-phe (pink), 4F-NAL-phe (teal), and 3,4,5F-NAL-phe (purple) with 0 μM PUFA (left) and 20 μM PUFA (right). (**C, E, G**) $I/I_0$, (**E**) $\Delta V_{0.5}$, and (**G**) $G_{max}$ dose–response curves

*Figure 3 continued on next page*

*Figure 3 continued*
for NAL-phe (n = 4), 4Br-NAL-phe (n = 3), 4F-NAL-phe (n = 4), and 3,4,5F-NAL-phe (n = 5) with dotted line representing dose response of NALT (n = 4). (**D, F, H**) Maximum effects on (**D**) $I/I_0$, (**F**) $\Delta V_{0.5}$, and (**H**) $G_{max}$ (at 20 μM) for 4Br-NAL-phe (n = 3), 4F-NAL-phe (n = 4), and 3,4,5F-NAL-phe (n = 5). Asterisks indicate statistically significant differences determined by one-way ANOVA with Tukey's test for multiple comparisons. Values for all compounds and concentrations available in *Figure 3—source data 1*.

The online version of this article includes the following source data for figure 3:

**Source data 1.** Source Data for Panels C-H.

$G_{max}$ increasing aromatic PUFA, on the $I_{Ks}$ channel with the mutation K326C to assess whether K326 is also important for the aromatic PUFA $G_{max}$ increasing mechanism (*Figure 5D*). The $G_{max}$ increasing effect of 3,4,5F NAL-phe was significantly decreased from 2.4 ± 0.4 at 20 μM in the WT $I_{Ks}$ channel to 1.22 ± 0.2 at 20 μM (*p=0.0287; *Figure 5E and F*). This reduction indicates that K326 is necessary for 3,4,5F NAL-phe's $G_{max}$ increasing effect.

To test whether the $V_{0.5}$ shift and $G_{max}$ increasing effects of aromatic PUFAs are independent of each other, we compare the Lin-tyr effects on both of the above mutations. The mutation R231Q+Q234R significantly decreases the $V_{0.5}$ shift effect from $\Delta V_{0.5}$ = –74.4 mV ± 4.1 at 20 μM in the WT $I_{Ks}$ channel to $\Delta V_{0.5}$ = –36.5 mV ± 7.3 at 20 μM in the mutation (****p < 0.0001), as mentioned above, but does not change $G_{max}$ increasing effect of Lin-tyr (*Figure 5G and H*). Similarly, the mutation K326C significantly decreases the $G_{max}$ effect of Lin-tyr from $G_{max}$ = 2.04 ± 0.59 at 20 μM in the WT $I_{Ks}$ channel to $G_{max}$ = 1.08 ± 0.15 at 20 μM in the mutation (***p=0.0015) but does not change the $V_{0.5}$ shift effect of Lin-tyr (*Figure 5G and H*). Overall, *Figure 5* demonstrates that aromatic PUFAs are shifting the voltage dependence and increasing the current maximum by two independent mechanisms: one via interactions with S4 charges (R231) and one via interactions with K326.

## Residue T224 in the S3–S4 loop is a novel locus for hydrogen bond formation between the $I_{KS}$ channel and tyrosine PUFAs

Our experiments using fluorinated NALT (NAL-3F-tyr) to improve the hydrogen bonding capacity of the tyrosine head group demonstrated that hydrogen bonding by the tyrosine's para-hydroxyl group is the reason for the large effect of PUFAs with tyrosine head groups on the $I_{Ks}$ channel voltage-dependent activation. To identify the residue with which the tyrosine head group hydrogen bonds, we mutated residues in the S3–S4 loop capable of hydrogen bond formation. We chose residues in the S3–S4 loop because it is located near the top of the PUFA head group in the proposed binding site for PUFAs to have the voltage-dependence effect. We individually mutated serine 217 (S217A), glutamine 220 (Q220L), threonine 224 (T224V), and serine 225 (S225A) and compared the effects of NALT on mutated channels compared to the WT $I_{Ks}$ channel (*Figure 6A and B*). We found that S217A, Q220L, and S225A showed similar maximum shifts in voltage-dependent activation compared to the wild-type channel (WT + NALT: $\Delta V_{0.5}$ = –56.1 ± 3.6 mV; S217A + NALT: –65.9 ± 3.7 mV; Q220L + NALT: –59.5 ± 11.1 mV; S225A + NALT: –52.4 ± 3.7 mV at 20 μM, ns; *Figure 6C and D*). However, the T224V mutation significantly attenuated the leftward shift in the voltage dependence of activation in response to NALT application from $\Delta V_{0.5}$ = –56.1 ± 3.6 mV in WT channels to $\Delta V_{0.5}$ = –32.1 ± 7.0 at 20 μM (*p=0.03; *Figure 6D*). To determine whether this effect was specific to compounds with the ability to form hydrogen bonds, we compared the effects of hydrogen-bonding NALT and non-hydrogen-bonding NAL-phe on T224V mutant channels (*Figure 6E*). In contrast to the attenuation of the overall voltage shift observed when NALT was applied to the T224V, there was no difference in the voltage-shifting effects of NAL-phe between the T224V mutant and WT channels (WT + NAL-phe:$\Delta V_{0.5}$ = –12.5 ± 3.8 mV; T224V + NAL-phe: –13.2 ± 2.6 mV at 20 μM, ns; *Figure 6F and G*). These data demonstrate that the T224V mutation only reduces the efficacy of aromatic PUFAs that contain a hydrogen-bonding group like tyrosine. As a result, we have identified a novel interaction between the S3–S4 loop residue T224 and hydrogen bonding moieties of aromatic PUFA head groups. *Figure 7A* shows the distance between the residues R231 and T224V in KCNQ1 is 9.1 Å and the length of a tyrosine head group is 8.5 Å. The tyrosine head group fits nicely in between these two residues in silico, demonstrating the possibility of this interaction to occur (*Figure 7A*).

In our proposed mechanism, aromatic PUFA analogues modulate the $I_{Ks}$ channel via two independent interactions (*Figure 5*). In one site, the carboxyl group is interacting with R231 (in S4) and

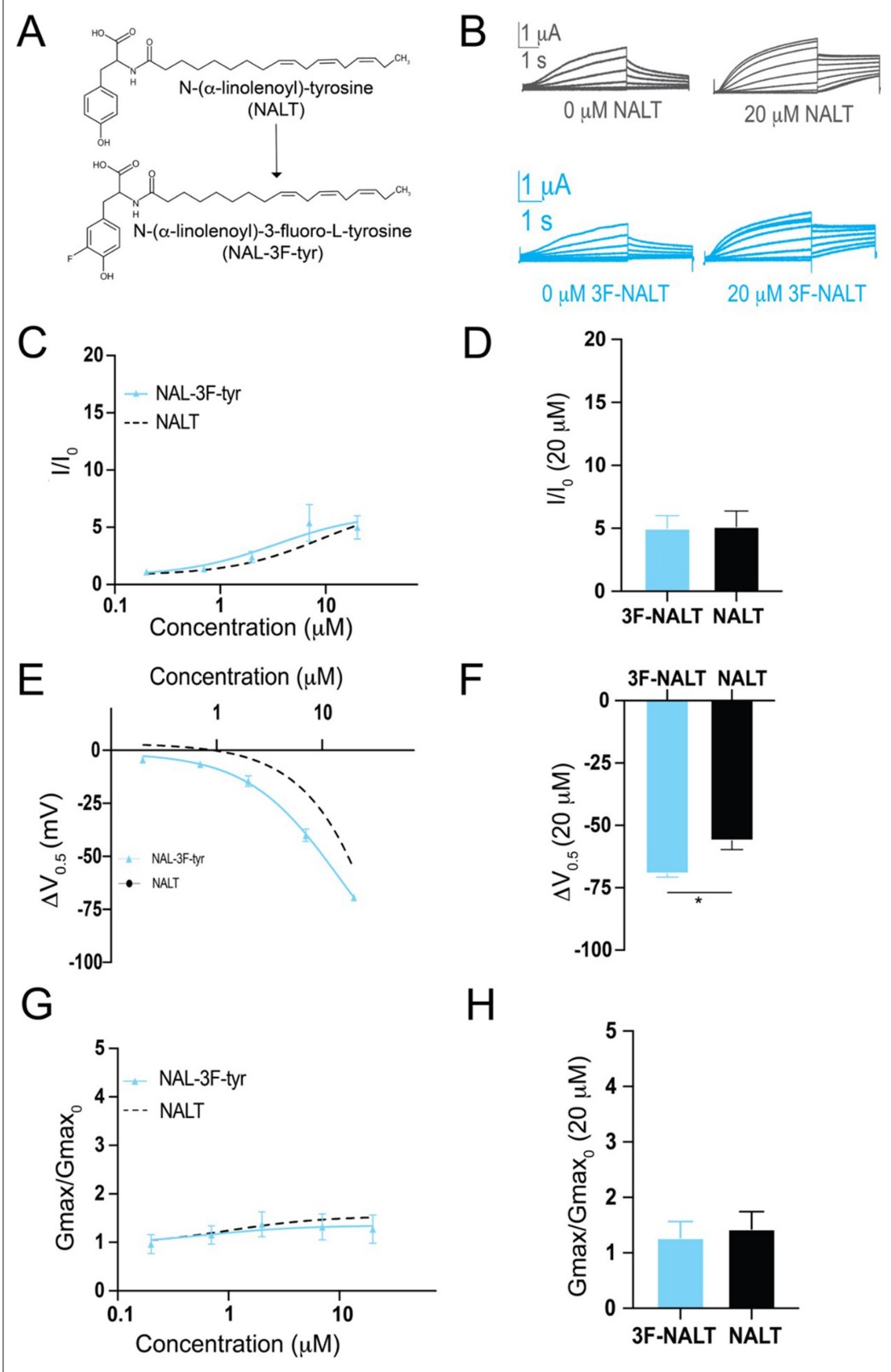

**Figure 4.** Hydrogen bonding through the distal –OH group of tyrosine polyunsaturated fatty acids (PUFAs) is important for effects on $I_{Ks}$ channel voltage dependence. (**A**) Structures of NALT and 3F-NALT. (**B**) Representative traces of NALT (gray) and 3F-NALT (cyan) with 0 μM PUFA (left) and 20 μM PUFA (right). (**C, E, G**) $I/I_0$, (**E**) $\Delta V_{0.5}$, and (**G**) $G_{max}$ dose–response curves for NALT (black dashed line) (n = 4) and 3F-NALT (cyan) (n = 3). (**D, F, H**) Maximum

*Figure 4 continued on next page*

*Figure 4 continued*

effects on (**D**) $I/I_0$, (**F**) $\Delta V_{0.5}$, and (**H**) $G_{max}$ (at 20 μM) for 3F-NALT (n = 3) and NALT (n = 4). Values for all compounds and concentrations available in *Figure 4—source data 1*.

The online version of this article includes the following source data for figure 4:

**Source data 1.** Source Data for Panels C-H.

the distal –OH is hydrogen bonding with T224 (in the S3–S4 loop) (*Figure 7B*). In the other site, the carboxyl group is interacting with K326 (in S6) and the distal electronegative atom stabilizes the PUFA via an electrostatic interaction (*Figure 7B*).

## Discussion

We have found that PUFA analogues with tyrosine head groups are strong activators of the cardiac $I_{Ks}$ channel. Tyrosine PUFAs shift the voltage dependence of activation to negative potentials and increase the maximal conductance, which together contribute to increases in overall $I_{Ks}$ current. The tyrosine head group is an aromatic ring with a distal -OH group in the para-position. Tyrosine PUFA analogues have the potential to interact with the $I_{Ks}$ channel through several candidate mechanisms involving either the aromatic ring or the -OH group (or both). The aromatic ring could modulate $I_{Ks}$ channel function through cation–pi interactions with positively charged groups on the $I_{Ks}$ channel. In addition, the -OH group could participate in electrostatic interactions and/or act as a hydrogen bond donor. In this work, we elucidate the mechanisms of this PUFA-induced activation of the $I_{Ks}$ channel by applying PUFA analogues with modified aromatic head groups designed to test specific chemical interactions between the PUFA head group and the $I_{Ks}$ channel.

If cation–pi interactions were the primary mechanism through which tyrosine PUFAs activate the $I_{Ks}$ channel, we would expect similar activating effects of PUFA analogues with aromatic rings that lack the –OH group, such as phenylalanine. However, PUFA analogues with phenylalanine head groups (Lin-phe and NAL-phe) do not activate the $I_{Ks}$ channel to the same degree as PUFA analogues with a tyrosine head group (Lin-tyr and NALT) and display significant reductions in efficacy for increases in $I/I_0$ and shifts in the $V_{0.5}$. Further evidence that cation–pi interactions are not a predominant mechanism for $I_{Ks}$ channel activation by tyrosine PUFA analogues comes from experiments applying fluorinated phenylalanine PUFAs (4F-NAL-phe and 3,4,5F-NAL-phe), which can be used as a tool to probe cation–pi interactions in ion channel function (*Pless et al., 2014*). *Pless et al., 2014* demonstrated that tri-fluorination of phenylalanine disperses the electrostatic surface potential that is necessary for cation–pi interactions (*Pless et al., 2014*). Disruption of the electrostatic surface potential through addition of fluorine atoms to the NAL-phe head group (3,4,5F-NAL-phe), therefore, is expected to reduce the efficacy of 3,4,5F-NAL-phe in comparison to NAL-phe alone. However, we find the opposite when we apply 3,4,5F-NAL-phe to the cardiac $I_{Ks}$ channel, and see that 3,4,5F-NAL-phe is a more potent activator of the $I_{Ks}$ channel compared to NAL-phe alone. Together, these data suggest that cation–pi interactions are not the primary mechanism through which these aromatic PUFA analogues activate the cardiac $I_{Ks}$ channel.

When we look at several fluorinated and brominated phenylalanine PUFA analogues, we find specifically that 3,4,5F-NAL-phe has significantly greater effects on $I/I_0$ and $\Delta V_{0.5}$ compared to NAL-phe alone. While not statistically significant, 4Br-, 4F-, and 3,4,5F-NAL-phe also lead to some of the most consistent increases in $G_{max}$ among the PUFA analogues tested in this work, with each of these compounds leading to a twofold increase in $G_{max}$. These data suggest that aromatic PUFA analogues with highly electronegative atoms on the distal end of the aromatic head group have the most pronounced effects on the maximal conductance of the $I_{Ks}$ channel. Although brominated and fluorinated phenylalanine analogues increase the maximal conductance of the $I_{Ks}$ channel, these modified PUFAs still fail to recapitulate the leftward $\Delta V_{0.5}$ observed with tyrosine PUFA analogues. While the –OH group of tyrosine PUFA analogues is indeed strongly electronegative, it can also act as a hydrogen bond donor. When we applied a fluorinated tyrosine PUFA (3F-NALT) to increase hydrogen bonding abilities, we found that this leads to a stronger leftward shift in the voltage dependence of $I_{Ks}$ activation. This suggests that hydrogen bonding via the –OH group contributes to the left-shifting effects of voltage-dependent activation through effects on the $I_{Ks}$ channel voltage sensor. Most notably, these results suggest that specific modifications to the aromatic PUFA head group can

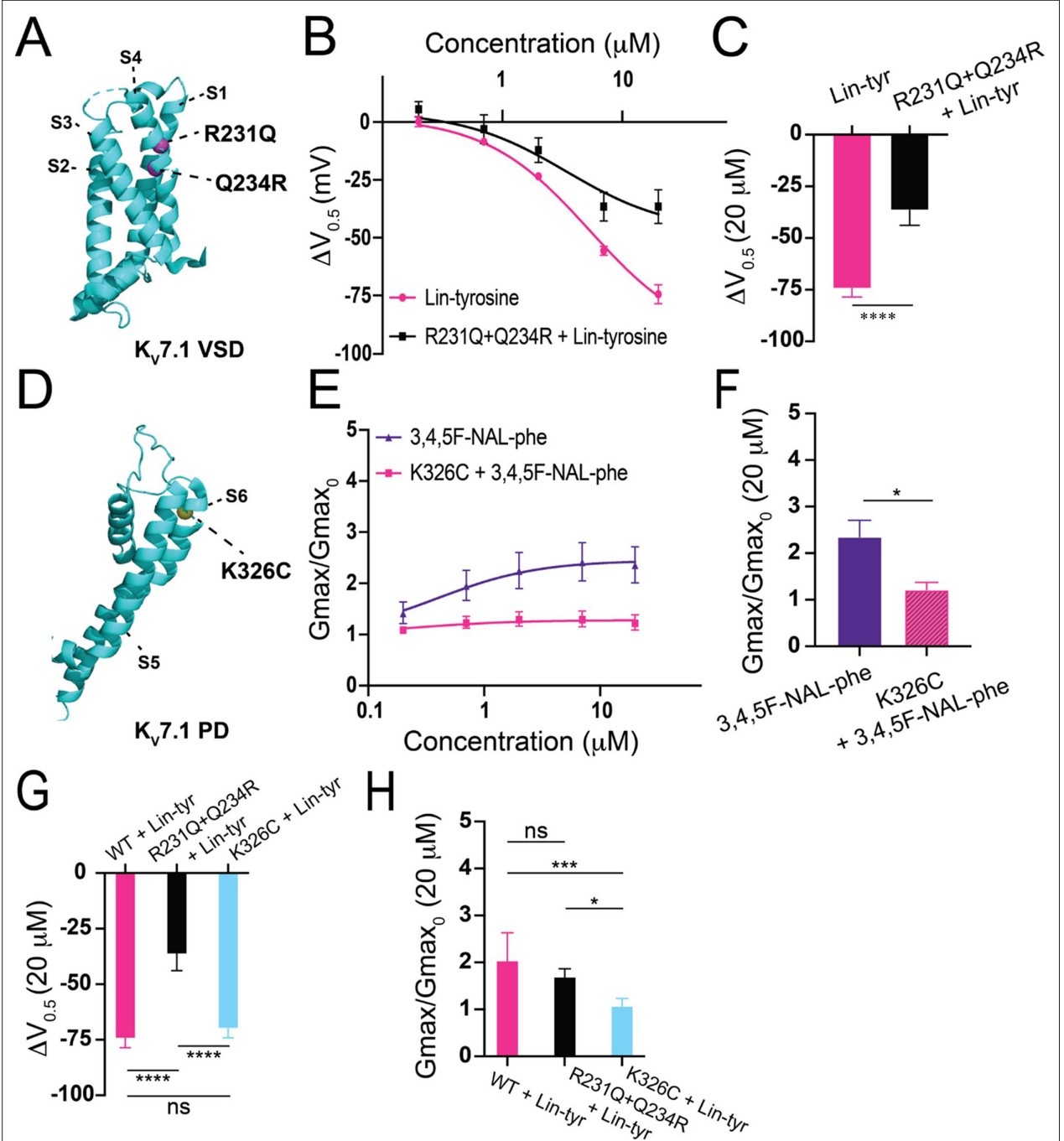

**Figure 5.** Proposed mechanisms of aromatic polyunsaturated fatty acids (PUFAs). (**A**) Structure of $K_V7.1$ voltage-sensing domain (VSD) based on PDB: 6V00A projected using PyMOL Software (Schrödinger L and DeLano 2020). Pink spheres indicate mutated residues in the S4 segment, R321Q-Q234R, which are implicated in PUFA-mediated effects on voltage dependent activation. (**B**) $\Delta V_{0.5}$ dose–response curve for WT $K_V7.1$/KCNE1 + Lin-tyr (pink) (n = 4) and $K_V7.1$-R231Q-Q234R/KCNE1 + Lin-tyr (black) (n = 5). (**C**) Maximum effects on $\Delta V_{0.5}$ (at 20 μM) for WT $K_V7.1$/KCNE1 + Lin-tyr (n = 4) and $K_V7.1$-R231Q-Q234R/KCNE1 + Lin-tyr (n = 5). (**D**) Structure of $K_V7.1$ pore domain (PD). Yellow spheres indicate mutated residue in the S6 segment, K326C, which is implicated in PUFA-mediated effects on maximal conductance. (**E**) $G_{max}$ dose–response curve for WT $K_V7.1$/KCNE1 + 3,4,5F-NAL-phe (purple) (n = 5) and $K_V7.1$-K326C/KCNE1 + 3,4,5F-NAL-phe (pink) (n = 3). (**F**) Maximum effects on $G_{max}$ (at 20 μM) for WT $K_V7.1$/KCNE1 + 3,4,5F-NAL-phe (n = 5) and $K_V7.1$-K326C/KCNE1 + 3,4,5F-NAL-phe (n = 3). (**G**) Maximum effects on $\Delta V_{0.5}$ (at 20 μM) of Lin-tyr on WT $K_V7.1$/KCNE1 (n = 4), $K_V7.1$-R231Q-Q234R/KCNE1 (n = 5), and $K_V7.1$-K326C/KCNE1 (n = 6). (**H**) Maximum effects on $G_{max}$ (at 20 μM) of Lin-tyr on WT $K_V7.1$/KCNE1 (n = 6), $K_V7.1$-R231Q-Q234R/KCNE1 (n = 5), and $K_V7.1$-K326C/KCNE1 (n = 6). Values for all compounds and concentrations available in *Figure 5—source data 1*.

The online version of this article includes the following source data for figure 5:

**Source data 1.** Source Data for Panels B, C, E, F, G, and H.

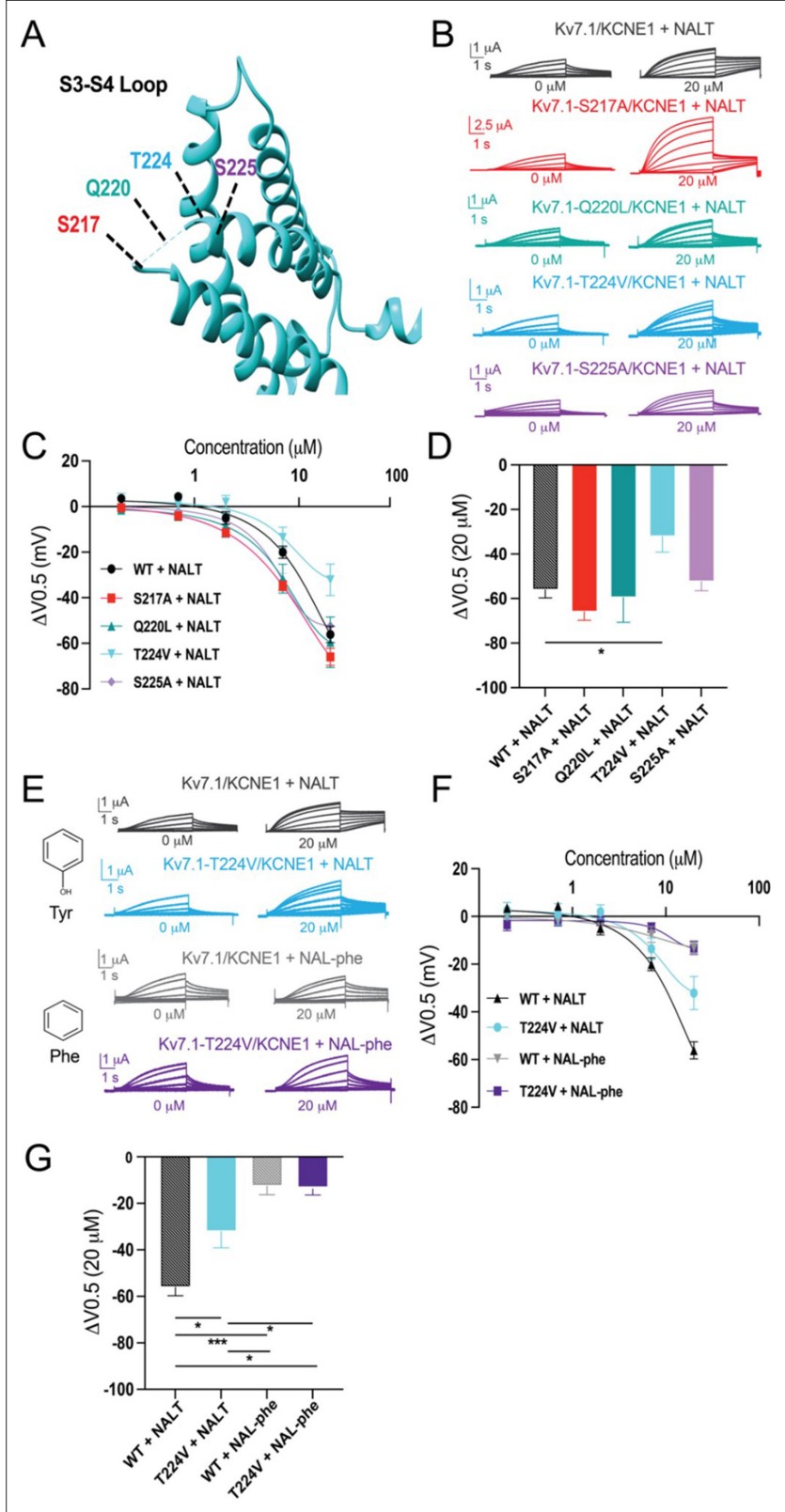

**Figure 6.** Loop is the locus for hydrogen bonding interactions with tyrosine polyunsaturated fatty acids (PUFAs). (**A**) Top view of $K_V7.1$ voltage-sensing domain (VSD) highlighting mutated residues in the S3–S4 loop. (**B**) Representative traces of WT $K_V7.1$/KCNE1 (black), $K_V7.1$-S217A/KCNE1 (red), $K_V7.1$-Q220L/KCNE1 (teal), $K_V7.1$-T224V/KCNE1 (cyan), and $K_V7.1$-S225A/KCNE1 (purple) with 0 µM (left) and 20 µM (right) NALT. (**C**) $\Delta V_{0.5}$ dose–

*Figure 6 continued on next page*

*Figure 6 continued*

response curve for WT $K_V7.1/KCNE1$ (n = 4), $K_V7.1$-S217A/KCNE1 (n = 5), $K_V7.1$-Q220L/KCNE1 (n = 3), $K_V7.1$-T224V/KCNE1 (n = 4), and $K_V7.1$-S225A/KCNE1 (n = 7) with NALT. (**D**) Maximum effects on $\Delta V_{0.5}$ (at 20 µM) for WT and S3–S4 loop mutations. Asterisks indicate statistically significant differences determined by one-way ANOVA. (**E**) Representative traces of WT $K_V7.1/KCNE1$ with NALT (black) and NAL-phe (gray) compared to $K_V7.1$-T224V/KCNE1 with NALT (cyan) and NAL-phe (dark purple), $K_V7.1$-S217A/KCNE1 (red), $K_V7.1$-Q220L/KCNE1 (teal), $K_V7.1$-T224V/KCNE1 (cyan), and Kv7.1-S225A/KCNE1 (purple) with 0 µM (left) and 20 µM (right) NALT. (**F**) $\Delta V_{0.5}$ dose–response curve for WT $K_V7.1/KCNE1$ and KV7.1-T224V/KCNE1 with NALT and NAL-phe. (**G**) Maximum effects on $\Delta V_{0.5}$ (at 20 µM) for WT $K_V7.1/KCNE1$ (n = 4) and $K_V7.1$-T224V/KCNE1 with NALT (n = 4) and NAL-phe (n = 7). Asterisks indicate statistically significant differences determined by one-way ANOVA. Values for all compounds and concentrations available in **Figure 6—source data 1**.

The online version of this article includes the following source data for figure 6:

**Source data 1.** Source Data for Panel C, D, F, and G.

preferentially improve either the voltage-shifting or maximal conductance effects of PUFA analogues. Our data suggests that adding highly electronegative groups to an aromatic ring, such as bromine and fluorine, most consistently improve the maximal conductance-increasing effects and reduce voltage dependence-shifting effects relative to PUFA analogues with a tyrosine or phenylalanine head group. On the other hand, we found that reducing the $pK_a$ of the –OH group (and increasing the potential for hydrogen bonding), while leaving the effect on $G_{max}$ intact, preferentially improves the voltage-shifting effects on the $I_{Ks}$ channel.

Previous work has demonstrated that PUFA analogues have two independent effects on $I_{Ks}$ channel activation. PUFA analogues are known to shift the voltage dependence of activation in the $I_{Ks}$ channel through electrostatic effects on the channel voltage sensor (**Börjesson et al., 2008**; **Liin et al., 2015**). This is mediated by interactions of the negative PUFA head group with the outermost positively charged arginine residues located in the S4 segment (**Liin et al., 2016**; **Liin et al., 2015**). Recently, though, a second effect on the $I_{Ks}$ channel pore has been reported to influence the maximal conductance of the $I_{Ks}$ channel (**Liin et al., 2018**). This is mediated through electrostatic interactions between the PUFA head groups and a positively charged lysine residue in the S6 segment – K326 (**Liin et al., 2018**). In addition, molecular dynamics (MD) simulations with the $K_v7.1$ (KCNQ1) channel (the pore-forming domain for the $I_{Ks}$ channel) (**Yazdi et al., 2021**) identified two separate high-occupancy sites for linoleic acid: site 1 at R231 in the S4 segment, and site 2 at K326 in the S6 segment (**Yazdi et al., 2021**). We here show that the more effective aromatic PUFAs also act on these sites in S4 and S6. To do this, we selected the best $V_{0.5}$ shifting aromatic PUFA (Lin-tyr) to test on the $I_{Ks}$ channel with the S4 mutation R231Q. Additionally, we selected the best $G_{max}$ increasing aromatic PUFA (3,4,5F-NAL-phe) to test on the $I_{Ks}$ channel with S6 mutation K326C. The mutation R231Q decreases the $V_{0.5}$ shifting effect of Lin-tyr by half, indicating that Lin-tyr is shifting the voltage dependence by creating an electrostatic interaction with the positive charges on the voltage sensor. Conversely, the mutation K326C almost completely removed the $G_{max}$ increasing effect of 3,4,5F-NAL-phe. To demonstrate that these effects are also independent of each other in aromatic PUFAs, we evaluate the $V_{0.5}$ shifting effect and the $G_{max}$ increasing effect of Lin-tyr on both mutations. The R231Q mutation decreases $V_{0.5}$ shifting effect but does not change the $G_{max}$ effect of Lin-tyr. While the K326 mutation decreases the $G_{max}$ effect but does not change the $V_{0.5}$ shifting effect of Lin-tyr. We therefore propose that the increased effects of the aromatic PUFAs, compared to non-aromatic PUFAs, are due to the additional hydrogen bonding in site 1 and electrostatic interactions in site 2 to better anchor them in these binding sites to increase their effects (**Figure 7B**). As mentioned above, we also show that the aromatic rings have the potential to be modified to give preferential effects on either the $I_{Ks}$ channel voltage sensor or channel pore.

Our experiments with NAL-Phe and 3F-NALT show that the hydrogen bonding capacity of the –OH on the tyrosine of NALT is necessary for it to have a more effective voltage dependence shift effect. We further discovered the specific details of the hydrogen bond interactions between this –OH group of NALT and the S3–S4 loop of the $I_{Ks}$ channel. We mutated all residues capable of hydrogen bonding in the S3–S4 loop, removing their ability to hydrogen bond and tested whether this changed the NALT voltage dependence shifting effect. The voltage dependence of mutations S217A, Q220L, and S225A was shifted to the same degree by NALT as the wild-type $I_{Ks}$ channel. However, the voltage

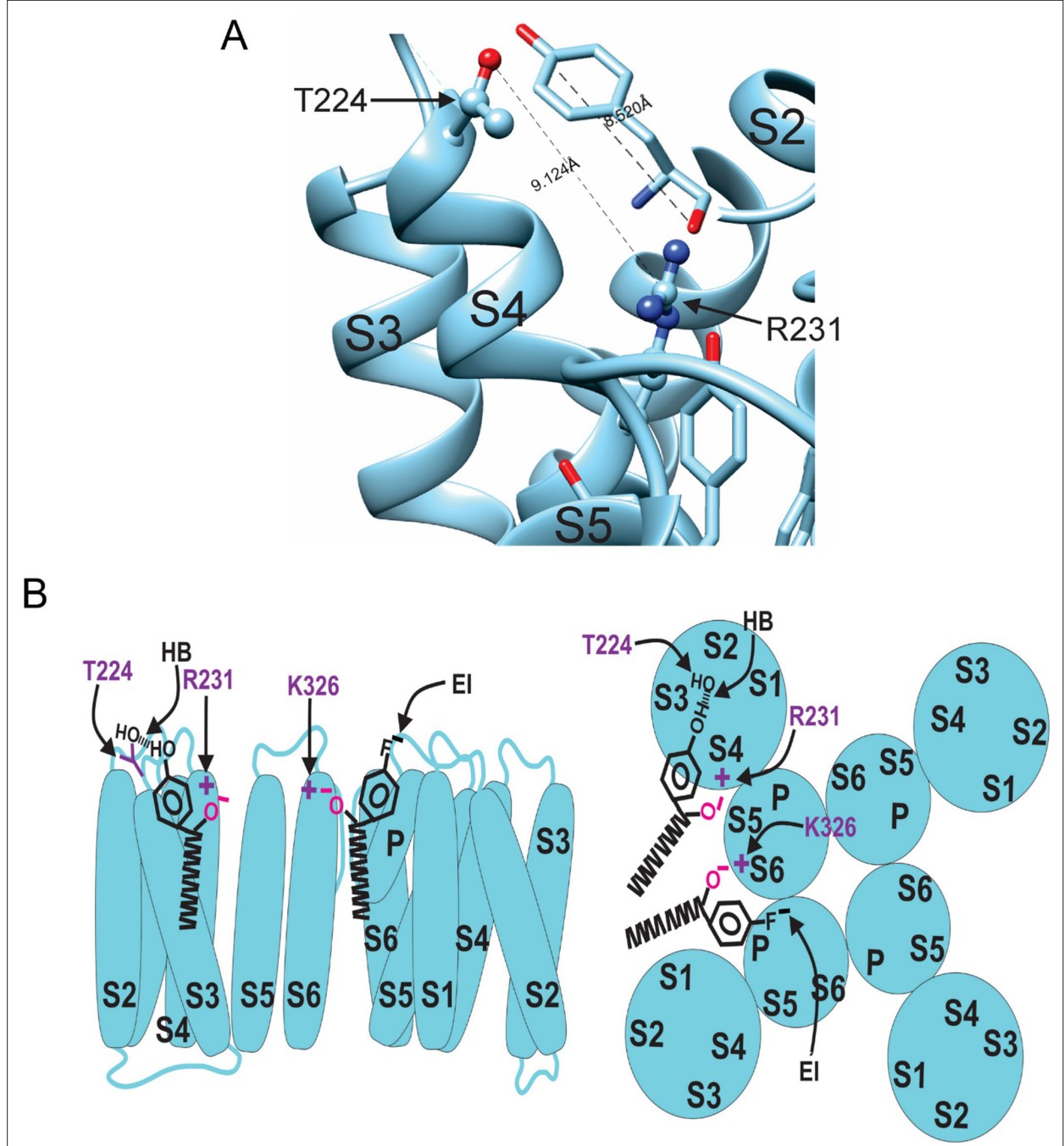

**Figure 7.** Model for aromatic polyunsaturated fatty acid (PUFA) interaction with $K_v7.1$/KCNE1 channels. (**A**) Image of the KCNQ1 structure (PDB 6UZZ) marking the distance between R231 and T224. This distance is shown in comparison to the length of a tyrosine head group. (**B**) Model for aromatic PUFAs effect on $K_v7.1$/KCNE1 channels, side view (left) and top view (right). One site is between S4 and S5: Aromatic PUFAs shift the voltage dependence of opening by stabilizing the upstate of S4 by an electrostatic interaction between R231(+) and the carboxyl group (O$^-$) of the PUFA. A hydrogen bond (HB) by the hydroxyl group (OH) at the para site of the aromatic ring of the PUFA with T224 stabilizes the PUFA in this site. Another site is between S6 and S1: aromatic PUFAs increase the maximum conductance by an electrostatic interaction between K326(+) and the carboxyl group (O$^-$). An electrostatic interaction (EI) by the para fluorine (F$^-$) stabilize the PUFA in this site.

dependence of mutation T224V was shifted significantly less than the WT $I_{Ks}$ channels. This shows that the –OH group on the tyrosine of NALT hydrogen bonds with T224V, thereby improving the PUFA's ability to shift the voltage dependence. This hydrogen bond interaction between PUFAs and the 3–4 loop of the $I_{Ks}$ channel is a novel mechanism to increase the effect of PUFAs to activate the $I_{Ks}$ channel.

These data suggest that the drugs designed to target this interaction would be more effective at shifting $I_{Ks}$ channel voltage dependence.

Overall, our findings suggest that different aromatic PUFA analogues not only increase PUFA efficacy on activating the $I_{Ks}$ channel, but their specific effects on $I_{Ks}$ function can be modulated independently, either increasing the maximal conductance or voltage-shifting effect. Patients with LQTS can have mutations in the $I_{Ks}$ channel that causes either a decrease in $G_{max}$ or a rightward shift of voltage dependence. By independently modulating PUFAs to either increase $G_{max}$ or cause a leftward shift in voltage dependence, we may be able to design therapies that are more personalized to each patient's specific LQTS mutation. This novel mechanistic understanding of how aromatic PUFAs have these effects on the $I_{Ks}$ channel may help to aid drug development for LQTS.

## Materials and methods
### Molecular biology

$K_V7.1$ and KCNE1 channel cRNA were transcribed using the mMessage mMachine T7 kit (Ambion). 50 ng of cRNA was injected at a 3:1, weight:weight ($K_V7.1$:KCNE1) ratio into defolliculated *X. laevis* oocytes (Ecocyte, Austin, TX) for $I_{Ks}$ channel expression. Injected cells were incubated for 72–96 hr in standard ND96 solution (96 mM NaCl, 2 mM KCl, 1 mM $MgCl_2$, 1.8 mM $CaCl_2$, 5 mM HEPES; pH = 7.5) containing 1 mM pyruvate at 16°C prior to electrophysiological recordings.

**Primers for $K_V7.1$ mutations**:

R231Q/Q234R – cggccatcaggggTatccAAttTctgAGAatcctgagAatg
K326C – ccagacgtgggtcgggTGCaccatcgcctcctgcttc
S225A – caggtgtttgccacgGCCgcTatcaggggTatccgcttcc
Q220L – gtgggctccaagggAcTTgtgtttgccacgtcgg
T224V – ggggcaggtgtttgcAGTgtcggcTatcaggggcatc
S217A – gtcctctgcgtgggcGccaaggggcaggtgtttg

### PUFA analogues and fluorinated PUFAs

Commercially available PUFAs N-(α-linolenoyl) tyrosine (NALT) item number 10032 and linoleoyl phenalanine (Lin-phe) item number 20063 were obtained from Cayman Chemical (Ann Arbor, MI) or Larodan (Solna, Sweden). Linoleoyl tyrosine (Lin-tyr), docosahexaenoyl tyrosine (DHA-tyr), and pinolenoyl tyrosine (Pin-tyr) were synthesized as described previously (*Bohannon et al., 2020*). NAL-phe, 4Br-NAL-phe, 4F-NAL-phe, 3,4,5F-NAL-phe, and 3F-NALT were synthesized similarly, with detailed descriptions of the synthesis procedures for each compound provided in the supplemental methods. PUFA analogues were kept at –20°C as 100 mM stock solutions in ethanol except 4Br-NAL-phe, 4F-NAL-phe, 3,4,5F-NAL-phe, and 3F-NALT where stock solutions were prepared as needed on the day of recording. Serial dilutions of the different PUFAs were prepared from stocks to make 0.2 µM, 0.7 µM, 2.0 µM, 7.0 µM, and 20 µM concentrations in ND96 solutions (pH = 7.5). NALT is a commercially available compound. The other compounds used in this study were synthesized in small amounts. Thus, NALT is used as the focus of this study.

### Two-electrode voltage-clamp (TEVC)

*X. laevis* oocytes, co-expressing wild-type $K_V7.1$ and KCNE1, were recorded in the two-electrode voltage-clamp (TEVC) configuration. Recording pipettes were filled with 3 M KCl. The recording chamber was filled with ND96 (96 mM NaCl, 2 mM KCl, 1 mM $MgCl_2$, 1.8 mM $CaCl_2$, 5 mM Tricine; pH 7.5). Dilutions of PUFAs and PUFA analogues were perfused into the recording chamber using the Rainin Dynamax Peristaltic Pump (Model RP-1) (Rainin Instrument Co., Oakland, CA). Electrophysiological recordings were obtained using Clampex 10.3 software (Axon, pClamp, Molecular Devices). During the application of PUFAs, the membrane potential was stepped every 30 s from –80 mV to 0 mV for 5 s before stepping to –40 mV and back to –80 mV to ensure that the PUFA effects on the current at 0 mV reached steady state (*Figure 1D*). A voltage-step protocol was used to measure the current vs. voltage (I–V) relationship before PUFA application and after the PUFA effects had reached steady state for each concentration of PUFA. Cells were held at –80 mV followed by a hyperpolarizing prepulse to –140 mV to make sure all channels are fully closed. The voltage was then stepped from

–100 to 60 mV (in 20 mV steps) followed by a subsequent voltage step to –20 mV to measure tail currents before returning to the –80 mV holding potential.

## Data analysis

Tail currents were analyzed using Clampfit 10.3 software in order to obtain conductance vs. voltage (G–V) curves to determine the voltage dependence of channel activation. $V_{0.5}$, the voltage at which half the maximal current occurs, was obtained by fitting the G–V curves from each concentration of PUFA with a Boltzmann equation:

$$G\left(V\right) = \frac{G_{max}}{1 + e^{(V_{0.5} - V)/s}}$$

where $G_{max}$ is the maximal conductance at positive voltages and $s$ is the slope factor in mV. The current values for each concentration at 0 mV ($I/I_0$) were used to plot the dose–response curves for each PUFA. These dose–response curves were fit using the Hill equation to obtain the $K_m$ value for each PUFA:

$$\frac{I}{I_0} = 1 + \frac{A}{1 + \frac{Km^n}{x^n}}$$

where $A$ is the fold increase in current caused by the PUFA at saturating concentrations, $K_m$ is the apparent affinity of the PUFA, $x$ is the concentration, and $n$ is the Hill coefficient. Fitted maximum values derived from the dose–response curves are reported for each of the effects ($I/I_0$, $\Delta V_{0.5}$, and $G_{max}$) from the different PUFAs tested. In some cases, there is variability in the $V_{0.5}$ between batches of oocytes. In order to correct for variability due to oocytes, when the $V_{0.5}$ was greatly different than 20 mV in control solution, we applied a correction in order to more accurately measure PUFA-induced $I_{Ks}$ current increases. We subtracted the $V_{0.5}$ (given by fitting the G–V with a Boltzmann equation) by 20 mV and used the current measured at the resulting voltage. The maximum conductance ($G_{max}$) was calculated by taking the difference between the maximum and minimum current values (using the G–V curve for each concentration) and then normalizing to the value $G_{max}0$ in control solution (0 µM). Graphs plotting mean and standard error of the mean (SEM) for $I/I_0$, $\Delta V_{0.5}$, $G_{max}$, and $K_m$ were generated using GraphPad Prism (GraphPad Software, La Jolla, CA).

## Statistics

Unpaired $t$-tests and one-way ANOVA with multiple-comparisons statistics were computed using GraphPad Prism (GraphPad Software). Results were considered significant if $p < 0.05$.

## Materials availability statement

Mutations and newly synthesized PUFAs are available from the corresponding author upon reasonable request.

## Acknowledgements

This work was supported by the Swedish Research Council (2021-01885 to SIL), the European Research Council (ERC) under the European Union's Horizon 2020 research and innovation program (grant agreement no. 850622 to SIL), and the National Institutes of Health (R01HL131461 to HPL). We thank Jason D Galpin and Christopher A Ahern (University of Iowa; R24 NS104617-04; the Facility for Atomic Mutagenesis) and Xiongyu Wu (Linkoping University) for synthesis of different aromatic PUFAs. DPT acknowledges support from the Canada Research Chairs program.

---

## Additional information

### Competing interests

Sara I Liin: A patent application (#62/032,739) including a description of the interaction of charged lipophilic compounds with the KCNQ1 channel has been submitted by the University of Miami with

HPL and SIL identified as inventors. H Peter Larsson: A patent application (#62/032,739) including a description of the interaction of charged lipophilic compounds with the KCNQ1 channel has been submitted by the University of Miamiwith HPL and SIL identified as inventors.Dr Hans Peter Larsson is the equity owner of VentricPharm, a company that operates in the same field of research as the study. The other authors declare that no competing interests exist.

### Funding

| Funder | Grant reference number | Author |
|---|---|---|
| HORIZON EUROPE European Research Council | 850622 | Sara I Liin |
| Swedish Research Council | 2021-01885 | Sara I Liin |
| National Institutes of Health | R01HL131461 | H Peter Larsson |

The funders had no role in study design, data collection and interpretation, or the decision to submit the work for publication.

### Author contributions

Briana M Bohannon, Conceptualization, Data curation, Formal analysis, Visualization, Writing – original draft, Writing – review and editing; Jessica J Jowais, Conceptualization, Data curation, Supervision, Investigation, Writing – original draft, Writing – review and editing; Leif Nyberg, Vanessa Olivier-Meo, Data curation; Valentina Corradi, D Peter Tieleman, Conceptualization, Writing – review and editing; Sara I Liin, Conceptualization, Resources, Data curation, Funding acquisition, Investigation, Project administration, Writing – review and editing; H Peter Larsson, Conceptualization, Formal analysis, Supervision, Funding acquisition, Validation, Visualization, Writing – original draft, Project administration

### Author ORCIDs

D Peter Tieleman ⓘ http://orcid.org/0000-0001-5507-0688
Sara I Liin ⓘ http://orcid.org/0000-0001-8493-0114
H Peter Larsson ⓘ http://orcid.org/0000-0002-1688-2525

### Decision letter and Author response

Decision letter https://doi.org/10.7554/eLife.85773.sa1
Author response https://doi.org/10.7554/eLife.85773.sa2

## Additional files

### Supplementary files

• MDAR checklist

### Data availability

All data generated or analysed during this study are included in the manuscript and supporting file; Source Data files have been provided for *Figures 1–6*.

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
