## [Editor Report]

This work reports important findings regarding the regulation of ion channels by polyunsaturated fatty acids (PUFAs) through the identification of novel aromatic PUFA analogs with potent effects on the IKs channels, which allow for mechanistic insights into their mode of action. The experiments are solid, combining site-directed mutagenesis, electrophysiological and pharmacological approaches to dissect the different molecular mechanisms and sites involved in the functional interactions. This work will be of broad interest to ion channel biophysicists, physiologists, and medical chemists interested in drug development for LQT syndrome.

---

## [Decision Letter]

**Decision letter after peer review:**

Thank you for submitting your article "Mechanistic insights into robust cardiac IKs potassium channel activation by aromatic polyunsaturated fatty acid analogues" for consideration by *eLife*. Your article has been reviewed by 3 peer reviewers, one of whom is a member of our Board of Reviewing Editors, and the evaluation has been overseen by Richard Aldrich as the Senior Editor. The following individual involved in the review of your submission has agreed to reveal their identity: Wayland WL Cheng (Reviewer #3).

Essential revisions:

All reviewers agreed that these results are solid and interesting. However, reviewers also raised several concerns that should be addressed by the authors. Essential revisions should include:

1) The choice of NALT as the focus of this study is not fully clear; please clarify.

2) You may wish to reconsider the conclusion that T224 forms a hydrogen bond with NALT or provide complementary evidence to support this claim (see full comments from reviewer #3).

3) Please clarify the rationale for the mutagenesis scan in the S3-S4 loop (reviewer #2).

4) The claim that aromatic PUFA analogues act through similar mechanisms to non-aromatic PUFAs would be strengthened by showing the (lack of/reduced) effects of aromatic PUFA analogues on conductance for the mutant R231Q/Q234R, or the (lack of/reduced) effects on deltaV0.5 for K326C mutants, as indicated by reviewers #1 and #3. In this context, it would be expected that the addition of Lin-Tyr, which causes significant effects on both Gmax and deltaV0.5, would present reduced effects in both K326C and R231Q/Q234R mutants.

5) We would recommend revising the text according to all comments raised by the reviewers and listed in the individual reviews below.

*Reviewer #1 (Recommendations for the authors):*

The abstract and introduction focus the aim of the work on the identification of mechanisms underlying the effect of N-a-linoleoyl Tyr (NALT) on Iks. However, four different Tyr PUFA analogues are actually used in this work, of which not NALT but Lin-Tyr has the most striking effect on Iks. It does produce a left shift of -74 mV and duplicates Gmax, resulting in an overall increase of 12.8 times IKs (I/Io). It would be interesting to explain why the focus is centered on studying NALT as opposed to Lin-Tyr, which seems to be more effective in increasing Iks.

As stated in this work and shown previously in other publications by the lab, it is convincingly shown that the Tyr PUFAs exert two different effects, by left-shifting the G-V and/or increasing channel conductance. It is clear that both effects can contribute synergistically to the net increase of IKs (I/Io). Coincident large effects in both (e.g. Lin-Tyr, -74.4 mV shift, and 2.0 Gmax increase) or modest effects of both (e.g. NAL-Phe, -13.1 mV shift, and 1.2 Gmax increase) result in large (12.5) or moderate (2.5) increase of I/Io. However, if one effect is predominant, it seems that the influence on IKs is more directly related to the increase in Gmax. For example, NALT produces a large shift of -56.2 mV and mild effects on the Gmax (1.43) which results in a 5.14 increase in IKs. Although a significantly larger shift is produced by 3F-NALT (-69.3) with similar effects on Gmax (1.3), however, this has no remarkable effect on IKs (5.0) as compared with NALT. When adding 3,4,5F-NAL-Phe, the effect on the G-V shift is modest (-32.4), but the increase in Gmax is significant (2.4), leading to higher net effects on Iks (7?) even than NALT. This may have been discussed somewhere else, but maybe the authors can comment on this apparent correlation. The higher effect of 3,4,5F-NAL-Phe on IKs would be worth discussing (Figure 3D)

Has the effect of Lin-Tyr on the K326C mutant been tested?

From previous works it is inferred that 3,4,5F-NAL-Phe on the K326C mutant is specific on Qmax, i.e. it produces a similar effect on V0.5 than WT, is that right? The same question would be asked for the effect of Lin-Tyr on Qmax on the R231Q/Q234R mutant.

About the model in Figure 5G: "An EI between para-F- and site 2 stabilize the PUFA in this site". Is this conclusion reached in the present work? It would be useful to include this more clearly in the text.

Could Figure 5G and 6H be integrated to provide an "updated model" of the Tyr-PUFA action mechanism? Figure 5G seems to recapitulate the previously described activation mechanisms of PUFAs on IKs channels (as stated in the text, line 342), which is probably already shown somewhere else. Maybe it would be more useful for the readers to include a single final model in Figure 6 where all findings are included. This is just a suggestion.

*Reviewer #2 (Recommendations for the authors):*

As a general comment, I would like to congratulate the authors on a manuscript well written, and results well-presented and illustrated; however, in my opinion, the manuscript could benefit from a little more introduction of why NALT could be of interest to reach a broader audience broaden the scope of interest of the manuscript. As stated beautifully in the introduction, expanding knowledge to include other "therapeutic interventions to treat LQTS" this manuscript could be an opportunity to highlight that outcome using tyrosine-PUFA analogs.

*Reviewer #3 (Recommendations for the authors):*

1) Regarding the effect of T224V, it appears that T224V produces an approximately 40% reduction in the deltaV0.5 for both NALT and NAL-phe, only that NAL-phe has a smaller effect and the standard deviation of the measurement means there is no significant difference between WT and T224V. This might suggest that T224V does not selectively perturb the effect of NALT compared to NAL-phe. The authors may wish to reconsider the conclusion that T224 forms a hydrogen bond with NALT or provide complementary evidence to support this claim.

2) Is it physically plausible from a structure of Kv7.1 that the hydroxyl in NALT should form a hydrogen bond with T224 when also interacting with the carboxylate group near R231? If so, it may be helpful to provide a molecular image that illustrates the proximity of these two residues.

3) Does R231Q/Q234R attenuate the effect of aromatic PUFA analogues on conductance? Likewise, does K326C attenuate the effect of aromatic PUFA analogues on deltaV0.5? As was previously reported for non-aromatic PUFAs, one might expect that these mutations will selectively impact just the voltage dependence of activation (R231) or conductance (K326). By showing this data, the authors would strengthen the claim that aromatic PUFA analogues act through similar mechanisms.

4) The authors state in lines 319 and 490 that the aromatic PUFAs are "superior". What is meant by this? It appears that the aromatic PUFAs are more efficacious than the non-aromatic analogues. If so, it would be helpful to state this more explicitly and to provide some comparison of the efficacy of aromatic and non-aromatic PUFAs.

5) In Methods line 117, Larsson et al., 2020, JGP is referenced. Which study is this in the Reference list?

---

## [Author Response]

Essential revisions:All reviewers agreed that these results are solid and interesting. However, reviewers also raised several concerns that should be addressed by the authors. Essential revisions should include:1) The choice of NALT as the focus of this study is not fully clear; please clarify.

We used NALT as our main compound because it is commercially available. The other PUFAs are all made by our chemists, and we get very small amounts in the synthesis. Therefore, we can do more experiments with NALT. We now mention this reason in the Methods section.

2) You may wish to reconsider the conclusion that T224 forms a hydrogen bond with NALT or provide complementary evidence to support this claim (see full comments from reviewer #3).

We have conducted more experiments on T224V with NAL-Phe. With this increase in sample size there is little to no difference in the NAL-phe effect on WT versus with the T224V mutation. This strengthens our conclusion that the OH group of NALT contributes to the ΔV0.5 effect via an interaction with T224.

3) Please clarify the rationale for the mutagenesis scan in the S3-S4 loop (reviewer #2).

The S3-S4 loop is close to the proposed binding site for PUFAs to have the DV0.5 effect. So, this was our first guess where potential hydrogen bonding partners would be located.

4) The claim that aromatic PUFA analogues act through similar mechanisms to non-aromatic PUFAs would be strengthened by showing the (lack of/reduced) effects of aromatic PUFA analogues on conductance for the mutant R231Q/Q234R, or the (lack of/reduced) effects on deltaV0.5 for K326C mutants, as indicated by reviewers #1 and #3. In this context, it would be expected that the addition of Lin-Tyr, which causes significant effects on both Gmax and deltaV0.5, would present reduced effects in both K326C and R231Q/Q234R mutants.

We now present data showing that the R231Q/Q234R mutation does not significantly reduce the Gmax effect and that the K326C mutation does not significantly reduced the DV0.5 effect on Lin-tyr

5) We would recommend revising the text according to all comments raised by the reviewers and listed in the individual reviews below.

We have responded to all the comments of the reviewers (see below).

Reviewer #1 (Recommendations for the authors):The abstract and introduction focus the aim of the work on the identification of mechanisms underlying the effect of N-a-linoleoyl Tyr (NALT) on Iks. However, four different Tyr PUFA analogues are actually used in this work, of which not NALT but Lin-Tyr has the most striking effect on Iks. It does produce a left shift of -74 mV and duplicates Gmax, resulting in an overall increase of 12.8 times IKs (I/Io). It would be interesting to explain why the focus is centered on studying NALT as opposed to Lin-Tyr, which seems to be more effective in increasing Iks.

We used NALT as our main compound because it is commercially available. The other PUFAs are all made by our chemists, and we get very small amounts in the synthesis. Therefore, we can do more experiments with NALT. We now mention this reason in the Methods section.

As stated in this work and shown previously in other publications by the lab, it is convincingly shown that the Tyr PUFAs exert two different effects, by left-shifting the G-V and/or increasing channel conductance. It is clear that both effects can contribute synergistically to the net increase of IKs (I/Io). Coincident large effects in both (e.g. Lin-Tyr, -74.4 mV shift, and 2.0 Gmax increase) or modest effects of both (e.g. NAL-Phe, -13.1 mV shift, and 1.2 Gmax increase) result in large (12.5) or moderate (2.5) increase of I/Io. However, if one effect is predominant, it seems that the influence on IKs is more directly related to the increase in Gmax. For example, NALT produces a large shift of -56.2 mV and mild effects on the Gmax (1.43) which results in a 5.14 increase in IKs. Although a significantly larger shift is produced by 3F-NALT (-69.3) with similar effects on Gmax (1.3), however, this has no remarkable effect on IKs (5.0) as compared with NALT. When adding 3,4,5F-NAL-Phe, the effect on the G-V shift is modest (-32.4), but the increase in Gmax is significant (2.4), leading to higher net effects on Iks (7?) even than NALT. This may have been discussed somewhere else, but maybe the authors can comment on this apparent correlation. The higher effect of 3,4,5F-NAL-Phe on IKs would be worth discussing (Figure 3D)

Note that all these experiments were made on wt channels, for which a shift in voltage dependence only have a minor effect on the maximum currents at 0 mV. This is due to that the V0.5 of wt channels is close to 0 mV. So, one cannot expect more than around a 2-3-fold increase in currents by a voltage shift. However, the idea is to use these compounds to treat LQTS and many LQTS mutations shift the voltage dependence of the IKs channels to more positive voltages. Therefore, these mutants have a low open probability at 0 mV and PUFAs have more room to increase the currents for these mutants by a voltage shift (>10-fold increase) (see Author response image 1).

**Author response image 1. sa2fig1:** 

Has the effect of Lin-Tyr on the K326C mutant been tested?

We now show data on LIN-Tyr on K326C.

From previous works it is inferred that 3,4,5F-NAL-Phe on the K326C mutant is specific on Qmax, i.e. it produces a similar effect on V0.5 than WT, is that right? The same question would be asked for the effect of Lin-Tyr on Qmax on the R231Q/Q234R mutant.

We now show data for that the R231Q/Q234R mutation does not significantly reduce the Gmax effect and that the K326C mutation does not significantly reduced the DV0.5 effect on Lin-tyr.

About the model in Figure 5G: "An EI between para-F- and site 2 stabilize the PUFA in this site". Is this conclusion reached in the present work? It would be useful to include this more clearly in the text.

Due to that replacement of the OH group by more negative groups (F- and Br-) made the aromatic PUFAs more effective in increasing Gmax, we proposed that an electrostatic effect cause the increase effect of para-F-NAL on Gmax. We now more clearly state our reasoning for this conclusion.

Could Figure 5G and 6H be integrated to provide an "updated model" of the Tyr-PUFA action mechanism? Figure 5G seems to recapitulate the previously described activation mechanisms of PUFAs on IKs channels (as stated in the text, line 342), which is probably already shown somewhere else. Maybe it would be more useful for the readers to include a single final model in Figure 6 where all findings are included. This is just a suggestion.

We now removed the two model figures in Figures5 and 6, and replaced them with one new one in Figure 7.

Reviewer #2 (Recommendations for the authors):As a general comment, I would like to congratulate the authors on a manuscript well written, and results well-presented and illustrated; however, in my opinion, the manuscript could benefit from a little more introduction of why NALT could be of interest to reach a broader audience broaden the scope of interest of the manuscript. As stated beautifully in the introduction, expanding knowledge to include other "therapeutic interventions to treat LQTS" this manuscript could be an opportunity to highlight that outcome using tyrosine-PUFA analogs.

We now added more about potential clinical application to the introduction to broaden the reader interest of the article.

Reviewer #3 (Recommendations for the authors):1) Regarding the effect of T224V, it appears that T224V produces an approximately 40% reduction in the deltaV0.5 for both NALT and NAL-phe, only that NAL-phe has a smaller effect and the standard deviation of the measurement means there is no significant difference between WT and T224V. This might suggest that T224V does not selectively perturb the effect of NALT compared to NAL-phe. The authors may wish to reconsider the conclusion that T224 forms a hydrogen bond with NALT or provide complementary evidence to support this claim.

We have conducted more experiments on T224V with NAL-Phe. With this increase in sample size there is little to no difference in the NAL-phe effect on WT versus with the T224V mutation. This strengthens our conclusion that the OH group of NALT contributes to the ΔV0.5 effect via an interaction with T224.

2) Is it physically plausible from a structure of Kv7.1 that the hydroxyl in NALT should form a hydrogen bond with T224 when also interacting with the carboxylate group near R231? If so, it may be helpful to provide a molecular image that illustrates the proximity of these two residues.

We now include a molecular model to show that NALT can interact with both R231 and T224 at the same time.

3) Does R231Q/Q234R attenuate the effect of aromatic PUFA analogues on conductance? Likewise, does K326C attenuate the effect of aromatic PUFA analogues on deltaV0.5? As was previously reported for non-aromatic PUFAs, one might expect that these mutations will selectively impact just the voltage dependence of activation (R231) or conductance (K326). By showing this data, the authors would strengthen the claim that aromatic PUFA analogues act through similar mechanisms.

We now show data for that the R231Q/Q234R mutation does not significantly reduce the Gmax effect and that the K326C mutation does not significantly reduced the DV0.5 effect in Lin-tyr.

4) The authors state in lines 319 and 490 that the aromatic PUFAs are "superior". What is meant by this? It appears that the aromatic PUFAs are more efficacious than the non-aromatic analogues. If so, it would be helpful to state this more explicitly and to provide some comparison of the efficacy of aromatic and non-aromatic PUFAs.

We now use efficacy instead of superior and make some comparison of efficacy.

5) In Methods line 117, Larsson et al., 2020, JGP is referenced. Which study is this in the Reference list?

We have fixed this reference.